# How Well Do Unsupervised Learning Algorithms Model Human Real-time and Life-long Learning?

**Chengxu Zhuang**[1,2]*, **Violet Xiang**[1]*, **Yoon Bai**[2] , **Xiaoxuan Jia**[3] , **Nicholas Turk-Browne**[4] ,
**Kenneth Norman**[5] , **James J. DiCarlo**[2] , **and Daniel L. K. Yamins**[1,6,7]

Department of Psychology[1], Computer Science[6],
and Wu Tsai Neurosciences Institute[7], Stanford University
[3]School of Life Sciences, Tsinghua University
[4]Department of Psychology and Wu Tsai Institute, Yale University
[5]Department of Psychology and Princeton Neuroscience Institute, Princeton University
[2]Department of Brain and Cognitive Sciences, MIT

*: Equal Contribution, Email: `chengxuz@mit.edu`

## Abstract

Humans learn from visual inputs at multiple timescales, both rapidly and flexibly acquiring visual knowledge over short periods, and robustly accumulating online learning progress over longer periods. Modeling these powerful learning capabilities is an important problem for computational visual cognitive science, and models that could replicate them would be of substantial utility in real-world computer vision settings. In this work, we establish benchmarks for both real-time and life-long continual visual learning. Our real-time learning benchmark measures a model's ability to match the rapid visual behavior changes of real humans over the course of minutes and hours, given a stream of visual inputs. Our life-long learning benchmark evaluates the performance of models in a purely online learning curriculum obtained directly from child visual experience over the course of years of development. We evaluate a spectrum of recent deep self-supervised visual learning algorithms on both benchmarks, finding that none of them perfectly match human performance, though some algorithms perform substantially better than others. Interestingly, algorithms embodying recent trends in self-supervised learning – including BYOL, SwAV and MAE – are substantially worse on our benchmarks than an earlier generation of self-supervised algorithms such as SimCLR and MoCo-v2. We present analysis indicating that the failure of these newer algorithms is primarily due to their inability to handle the kind of sparse low-diversity datastreams that naturally arise in the real world, and that actively leveraging memory through negative sampling – a mechanism eschewed by these newer algorithms – appears useful for facilitating learning in such low-diversity environments. We also illustrate a complementarity between the short and long timescales in the two benchmarks, showing how requiring a single learning algorithm to be locally context-sensitive enough to match real-time learning changes while stable enough to avoid catastrophic forgetting over the long term induces a trade-off that human-like algorithms may have to straddle. Taken together, our benchmarks establish a quantitative way to directly compare learning between neural networks models and human learners, show how choices in the mechanism by which such algorithms handle sample comparison and memory strongly impact their ability to match human learning abilities, and expose an open problem space for identifying more flexible and robust visual self-supervision algorithms.

36th Conference on Neural Information Processing Systems (NeurIPS 2022) Track on Datasets and Benchmarks.

# 1   Introduction

Deep neural networks (DNNs) optimized to perform visual recognition tasks using a large-scale human labeled dataset – ImageNet [15] – have produced state-of-the-art visual models [39, 50, 26]. Moreover, they have also been the most quantitatively accurate predictive models of neuronal responses in different sensory areas in the primate brain [57, 31, 5]. Their behavioral error patterns are also more consistent with those of non-human primates and humans than alternative models [49]. However, these models are biologically implausible due to the requirement for substantial human-annotated labels during training, which are extremely costly, if not impossible, for real organisms to obtain. Recently, unsupervised learning models have made significant progress in closing the gap to supervised models in performance on visual recognition tasks without the need for labeled data [56, 62, 52, 27, 12, 10, 22, 11, 58, 7, 9, 28]. Comparisons of these models to neuronal data in Zhuang et al. [64] and Konkle and Alvarez [38] show that they achieve high neural predictivity in early, middle, and higher cortical areas of the ventral visual stream (VVS). Even when these algorithms are trained on noisy and limited first-person videos collected from head-mounted cameras on three infants [51], these algorithms still yield competitive neural predictivity [64] and reasonable performance on small-scale categorization tasks [46].

However, these new powerful unsupervised algorithms have the potential to go beyond just the ability to achieve high performance or, post-training, match the static adult human representation – which supervised models already do reasonably well. Indeed, because these models can leverage the unlabelled stimuli used by biological organisms during visual learning, it is plausible that they might describe the *learning dynamics* of human behaviors under all time-scales. A model that had this capacity would be of great value both for understanding the biological mechanisms underlying visual development [33, 2, 41], as well as solving continual learning challenges in computer vision and robotics [42, 14, 25, 45].

In this work, we propose benchmarks for both real-time and life-long visual learning. Our real-time learning benchmark is constructed through quantifying the error in matching the visual categorization behavior changes in human adults reported by Jia et al. [30] (MIT License) during hour-long sessions. Our life-long learning benchmark is built using SAYCam [51] (License CC-BY 4.0) to create a training curriculum based on the visual diet experienced by human children over several years, with data presented in the same order and roughly the same duration as how the children experienced them. We then train DNNs using this naturalistic curriculum. Critically, this use of SAYCam differs from recent work such as Orhan et al. [46] and Zhuang et al. [64], where the video clips are simply used with a standard offline training protocol involving randomization and batching, which fails to capture the temporal structure of how experiences accrete over time in children. These two benchmarks are naturally complementary, because requiring a single learning algorithm to be locally context sensitive enough to match real-time learning changes while stable enough to avoid catastrophic forgetting over the long term is a very strong constraint.

Within this framework, we evaluate multiple high-performing unsupervised learning algorithms. Surprisingly, we find that several of the more recently proposed self-supervised algorithms, including BYOL [22], SimSiam [11], SwAV [7] and MAE [28], largely fail to match human learning in the real-time benchmark and show lower performance in the life-long benchmark, compared to an earlier generation of algorithms like SimCLR [10] and MoCo v2 [27, 12]. We find that the best-performing algorithms on both benchmarks share a key algorithmic design feature: actively contrasting one example with another, a way of leveraging memory called *negative sampling* that has been actively avoided in more recent algorithmic approaches. To test whether this design indeed facilitates learning in a low-diversity environment, we create an algorithm variant of BYOL through adding negative sampling and show that this variant greatly outperforms vanilla BYOL on both our short and long-term learning metrics. We also add this design to DINO [9], a high-performing ViT-based contrastive learning algorithm, and find that it consistently improves performance in the life-long benchmark.

Additionally, we systematically investigate how key parameters of the continual learning process influence performance for the two benchmarks and identify an underlying trade-off between them that acts as a strong constraint on human-like learning models. Finally, we perform an analysis indicating that one major mechanism underlying poor performance on our real-time learning benchmark is an algorithm's inability to capture the sparse learning signals in low-diversity (but natural) environments.

In the following sections, we first review relevant literature in Sec. 2. Then, we describe methods including how the benchmarks are constructed and how the continual learning process is constructed in Sec. 3. Following the method section, we show the results and the analyses in Sec. 4. Finally, we discuss limitations and future directions in Sec. 5.

## 2   Literature Review

*Unsupervised Learning Algorithms.* Recent progress in contrastive learning models has significantly improved performance on standard ImageNet benchmark, closing the gap between unsupervised and supervised models [22, 56, 10, 27, 12, 11, 58, 8, 62, 52] and neural predictivity [64, 38]. A subset of these models explicitly sample negative embeddings from different places including a memory bank [56], a memory queue [27, 12], and other input images from current batch [10]. Recent efforts removing negative samples have produced state-of-the-art performance [11, 22, 58, 8]. However, even without negative samples, these algorithms may rely on batch normalization to implicitly contrast embeddings of positive pairs with embeddings of other pairs in the same batch [53]. More recently, contrastive learning algorithms have also been used to train ViTs [16] and shown good performance [9, 13]. Additionally, a masked autoencoding objective has been proposed and proven efficient in training large-scale ViTs [28], which opens space for an entirely different route for unsupervised DNN training than contrastive learning algorithms. It is therefore interesting to evaluate whether models trained by this different algorithm perform in a human-like fashion.

*Real-time and Continual Visual Learning in Real Organisms.* Jia et al. [30] reported human visual categorization performance changes after unsupervised visual experience. Conceptually similar effects have also been found in individual primate IT neurons [43]. These effects are also potentially the neuronal basis for the behavioral changes of human subjects [30]. As for continual learning at a longer scale, early cortical organization is considerably mature at birth [55, 18], but the development of higher cortical areas and their processes underlying global form perception is a matter of ongoing debate [34, 35]. Although monkeys and humans can perceive elementary contours and discriminate textures quite early [1, 17], the ability to perceive composite patterns built from contours and texture takes much longer (2-3 years), reminiscent of that for global motion perception [17, 36].

*Unsupervised Deep Neural Network Models for the Visual System.* DNNs trained with contrastive learning algorithms on ImageNet have been shown to accurately predict the neural responses from multiple cortical areas of VVS [64, 38]. Apart from contrastive learning algorithms, Higgins et al. [29] show $\beta-$VAE, optimized to reconstruct the input image and simultaneously encode semantically meaningful hidden variables, can discover important factors for faces in a similar way as macaque IT neurons. However, it is unclear whether $\beta-$VAE produces quantitatively similar responses towards general stimuli as the neural responses from the VVS. Although these unsupervised learning algorithms yield accurate models of the visual system, they have not been used to model the specific patterns of learning dynamics in the visual system. Moreover, the training curriculum in prior work repeatedly presents the whole training dataset in a standard offline batched fashion, breaking the temporal structure of natural experience. In this work, we address both issues by testing the unsupervised DNNs on both the real-time and the life-long learning benchmark.

*Curriculum and Life-long Learning for Neural Networks.* Research in curriculum learning aims to develop specific curricula to improve training efficiency [61, 3, 32, 21, 24, 60]. In contrast, here curriculum structure is not a free variable: we work with (as natural as possible an approximation of) the actual curriculum of child learning to evaluate and improve algorithms. Networks trained on our realistic learning curriculum perform worse than networks using the offline curriculum, possibly due to catastrophic forgetting. Solving this issue is a major focus in life-long learning for neural networks [47]. Although this issue can be resolved through accumulating the learning experiences in a "memory" storage and jointly learning from memory and the current context, maintaining this continually-growing storage will be undesirable in many real-world applications. Therefore, methods like Elastic Weight Consolidation [37] and Generative Replay [54] have been proposed to address this issue without the need to maintain the storage, though these methods still underperform the storage solution. However, these methods are typically developed for training curriculum with drastic task or domain shifts, which is different from the life-long curriculum where no explicit tasks are defined and the domain shifts more smoothly. So in this work, we adapt the memory-storage solution and further explore how mixing it with the current-context learning with different ratios influences performance.

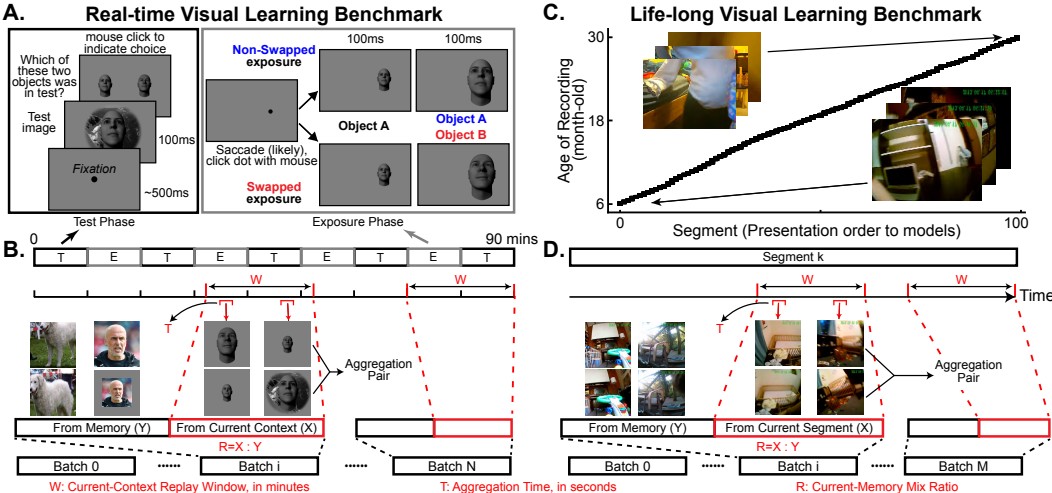

Figure 1: **Real-time and life-long visual learning benchmarks. A.** In the real-time benchmark, test and exposure phases are iterated for both humans and DNNs to correspondingly measure the object discrimination performance and present pairs of objects selected based on the experiment condition ("Swapped" or "Non-Swapped"). The schema for humans is provided in this panel as an example. For the Swapped condition, exposure phases show subjects or DNNs different-sized images from different objects, while for the Non-Swapped condition, the images are from the same object but with different sizes. **B.** Models learn from the whole datastream including both test and exposure phases, each of which takes 10 minutes. Learning is done in batches, where each batch consists of two parts: one part sampled from memory and the other part sampled from a sliding time window containing the recent visual experience in the current context, whose length is called the current-context replay window. The ratio between these two parts is called the current-memory mix ratio. Each item in one batch aggregates two temporally nearby images randomly sampled from a short time window called the aggregation time. **C.** In the life-long benchmark, models sequentially learn from the first-person infant videos in the SAYCam dataset grouped in segments and sorted by the age of the infant when these videos were recorded. **D.** Similar to B, models evaluated in the life-long benchmark jointly learn from memory (previous segments) and the current segment.

## 3 Methods

*Real-time Learning Benchmark.* This benchmark tests the models on five test phases separated by four exposure phases, following how humans were tested in Jia et al. [30] (Fig 1 **A**). To test the models, we first constructed a visual stimuli stream through simulating what humans were perceiving during their experiments. For example, the corresponding part of this stream for one exposure phase was built through concatenating the approximated visual stimulus of 400 exposure trials. Each trial contained 200ms presentation of the two object images followed by the gray background images for 1300ms (see **SI** Fig 5 **A** for examples). The gray background images serve as a proxy for the visual inputs of human subjects during inter-trial intervals. All stimulus are grayscale images, as Jia et al. [30] tested human subjects with grayscale images. The stimuli stream for one test phase was constructed through simulating 200 test trials. In the test trial for human subjects, one test image that was created by placing a big or small sized object in front of a randomly selected background was first presented after the 500ms fixation time. This test image was only presented for 100ms and followed by the image of the two middle sized objects put together. The human subjects would then be required to make a choice between the two objects before moving on to the next trial. To approximate the visual stimulus humans perceive during one test trial, we built the stream for the test trial through starting from the gray background image for 500ms. It was then followed by the test image for 100ms. We further hypothesized that human subjects made saccades between the two objects after the test image and simulated four saccades across the two presented middle sized objects, of which the interval was 600ms. Specifically, the test image was immediately followed by four blocks of single object images, each of which contained 600ms presentation of one of the two object images. We provide a more detailed pseudo-code description of this stream construction process in the Supplementary Information (**SI**, see Alg 1 and Sec. 1.1.2 in **SI**). Although this process involves several key parameters which were conveniently set as constants, such as the number of saccades and

the interval of two saccades, we have verified that reasonably varying these constants or changing them to be stochastic does not change our conclusions.

After constructing the visual stimuli stream, we then sampled from this stream to get batches of images that were fed to the DNNs to train them. This sampling procedure is described later in this section as the continual learning process. The DNNs homogeneously learn from their perceived visual input, regardless of whether it was from test or exposure phases. DNN outputs during test phases were extracted to compute the categorization performance (measured using $d'$) and then to compute the learning effects through subtracting the changes of $d'$ on the exposed objects by the changes of $d'$ on the non-exposed objects (see **SI** Sec. 1.1.3 for details). These learning effects are then compared to the human data collected for all the three experiment conditions (Non-Swapped, Swapped, and Switch conditions). The Non-Swapped and the Swapped conditions correspondingly keep or change the object identities in the two images (Fig 1 **A**), in which humans show increasingly better or worse categorization performance. The Switch condition combines the first two exposure phases of the Non-Swapped condition and the later two exposure phases of the Swapped condition, which therefore leads to first increasing and then decreasing human learning effects. For one test phase of one condition, the absolute difference between the model effects and human effects is computed and then averaged across all bootstrapping samples. This difference is then normalized by the same measure from the mean of human effects, making its minimal value 1 (see **SI** Sec. 1.1.4). Because the result from the first test phase, which is before the exposure phase, is used as a baseline in the learning effect computation (see **SI** Sec. 1.1.3), only the learning effects from the later four test phases are meaningful. As there are three conditions, the difference across all these 12 phases is averaged to get the final mismatch score to human. In addition to this aggregated mismatch score across all test phases, all of the bootstrapped values of the per-test difference score are also compared to 1 to measure the statistical significance of this individual score being different from 1. Also, we find that the initial $d'$ on these tested objects (faces in particular) is important for matching the human learning effects (see **SI** Sec. 1.1.6). Therefore, we pre-train the models on both ImageNet and VGGFace2 [6] * with a gray-filled random-central-positioned data augmentation added to the original data augmentation pipeline (see Fig 1 **B** for examples and **SI** Sec. 1.1.1). We fix the number of total updates for the models (150 steps each phase) but allow a freely-moving learning rate to get the minimal mismatch score (see **SI** Sec. 1.1.5 for more discussion of this).

*Life-Long Learning Benchmark.* We first create a subset of SAYCam by taking all videos from child Sam, yielding 200 hours of videos, called SamCam. These videos are then sorted by the age they were taken and then grouped into 100 segments, which are sequentially presented to the models (Fig 1 **C**). The models trained on these segments are evaluated every 10 segments through extracting their features on a subsampled ImageNet (MiniImageNet) and testing the performance using SVM (see **SI** Sec. 1.2.2). All 10 performance numbers are averaged to get the final measure, which is called the "trajectory-averaged Mini-ImageNet performance".

*Continual Learning Process.* Intuitively, three factors characterize continual visual learning: how learning from memory and the current context are mixed, how much of the recent visual experience in the current context is replayed, and how temporally close two visual stimulus need to be to get aggregated. For example, more learning from memory means better long-term learning performance but potentially less flexible in real time as that leads to less focus on the current context. Similarly, sampling from a longer replay time window in the current context with a fixed budget enables the simultaneous learning or contrasting of more diverse visual experiences but also risks in missing the very recent learning signals as less of them are sampled.

We formalize these factors in both benchmarks via parameterizing a standardized continual learning process, in which models learn from batches constructed through mixing samples from memory and a recent time window in the current context. The memory in the real-time benchmark is the pre-training dataset (ImageNet and VGGFace2), whereas the memory in the life-long benchmark is the previous segments. To get the part from the current context, a time point corresponding to each batch is first computed depending on its relative position in the whole segment. For example, the time point for the last batch in the real-time benchmark is 90 minutes, while that in the life-long benchmark is the end of the current segment (Fig 1 **B, D**). This time point is then the end point of the replay window whose length is controlled by the current-context replay window ($W$), from which the

---

*Although this dataset has been taken offline, this pretraining process should also work with other large-scale face datasets such as CelebA dataset [44], since the face test images are quite general and independent of VGGFace2 (see Fig 1 **A**).

visual experience is sampled to form the current context. To get the samples, a short time window of length aggregation time (T) is first sampled within the replay window. Two images are then randomly sampled within this short window as the inputs to the models (Fig 1 **B, D**). Finally, the ratio between the samples from memory and the current context is controlled by the current-memory mix ratio (R). See **SI** Alg. 2 and Alg. 3 for pseudo-code descriptions of this process in the real-time and life-long learning benchmarks.

*Unsupervised Learning Algorithms.* In general, contrastive learning algorithms use DNNs to project high-dimensional raw pixel inputs into a lower-dimensional compact space and optimize the DNNs to make embeddings "robust" to data augmentation. Specifically, let $f$ represent the DNN being optimized and $x$ represents an arbitrary input image, contrastive learning algorithms first sample two data augmentations ($v^0$ and $v^1$) and then optimize $f$ to have two resulting embeddings ($e^0 = f(v^0(x))$ and $e^1 = f(v^1(x))$ in dimension $D$) be predictive of each other. Since both the real-time and the life-long benchmarks require the models to learn from the temporal statistics in videos, we follow the practice introduced in Zhuang et al. [63] to aggregate the embeddings of two images ($x_0$ and $x_1$) sampled from a short time window, meaning that $e^0 = f(v^0(x_0))$ and $e^1 = f(v^1(x_1))$. This work benchmarks the following algorithms: SimCLR [10], MoCo v2 [12], BYOL [22], SimSiam [11], Barlow-Twins [58], SwAV [8], DINO [9], and MAE [28]. Because they are all previously published algorithms, we only briefly describe them here. SimCLR treats a batch of input images as a group and uses other images in the same group as negative samples to be separated from both $e^0$ and $e^1$. MoCo v2 also uses negative samples, but it samples them from a maintained queue of recent embeddings. Another difference between SimCLR and MoCo v2 is that MoCo v2 maintains a running average of the optimized DNN as the target network, also called "momentum encoder" ($\hat{f}$). So $e^1$ is replaced with $\hat{f}(v^1(x_1))$. BYOL also uses $\hat{f}$, but it does not use negative samples. Instead, it only tries to predict $e^1$ from $e^0$ using a Multi-Layer-Perceptron (MLP). SimSiam is like BYOL without momentum encoder and with stop gradient operation on the target embeddings. SwAV maintains trainable prototypes and optimizes $f$ to achieve identical assignments of $e^0$ and $e^1$ to these prototypes. Barlow-Twins is like "transposed" SimCLR. SimCLR maximizes the diagonal elements and minimizes the off-diagonal elements of the matrix $E^0 E^{1^T}$, where $E^0$ and $E^1$ are batched $e^0$ and $e^1$ in the shape of (bs, $D$) (bs is batch size). Barlow-Twins does the same thing, but to $E^{0^T} E^1$. DINO is similar to BYOL on ViTs, but with additional practices including softmaxing $e^0$ and $e^1$ and centering $e^1$. MAE randomly masks out patches of $v^0(x_0)$ and then uses ViTs to reconstruct the masked patches. In our benchmarks, the target of MAE is changed to the masked patches of $v^0(x_1)$. We additionally create two variants through introducing SimCLR-style negative sample choice and loss definition to BYOL and DINO, called BYOLNeg and DINONeg (see **SI** Sec. 1.2.3).

Our implementations are based on OpenSelfSup [59]. For most of the algorithms, we use ResNet-18. For algorithms using ViTs, we use ViT-S. We additionally test SimCLR-ResNet-50 as Resnet-50 has a similar number of trainable parameters as ViT-S. The code used for our paper and the trained model checkpoints are in **SI** and `https://github.com/neuroailab/VisualLearningBenchmarks`. More details are in **SI** Sec. 1.2.1.

## 4   Results

*Life-Long Learning Results.* We first systematically vary the current-context replay window and the current-memory mix ratio to show how these two parameters influence the results on the life-long benchmark (Fig 2). Although these two parameters only control within-batch diversity, they significantly influence the life-long results. Specifically, for the given algorithm, its performance consistently improves whenever the change of the parameters increases within-batch diversity (also see **SI** Fig 1). Although all algorithms show this consistent change with respect to the within-batch diversity, the magnitude of this change greatly differs across algorithms. In fact, the performance of algorithms without negative sampling, including SwAV, BYOL, and SimSiam, is much worse than SimCLR, MoCo v2, Barlow-Twins, and BYOLNeg in the medium-diversity condition (short replay window with balanced mix ratio), and catastrophically fails in the low-diversity condition (short replay window with more current learning). Even in the high diversity condition, BYOL and SwAV perform worse compared to SimCLR, unlike the result on ImageNet, where both previous reports and our reimplementation find that SwAV and BYOL significantly outperform SimCLR (see **SI** Fig 4). This inconsistency can actually be explained by the higher sensitivity of SwAV and BYOL to the within-batch diversity compared to SimCLR, as SamCam is in general less diverse compared to

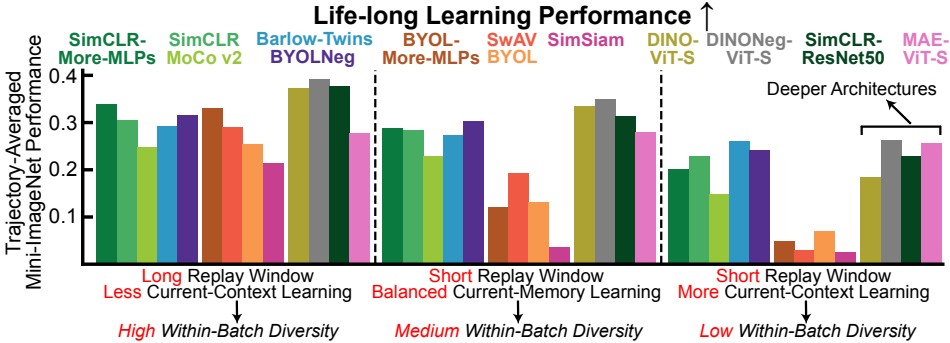

Figure 2: **Life-Long Learning results.** Life-long benchmark performance measured by the trajectory-averaged Mini-ImageNet performance. Three evaluated continual learning conditions are shown here with different current-context replay windows and current-memory mix ratios. Long replay window means $W = 20m$ and short window means $W = 0.5m$. More current-context learning means $R = 3:1$, balanced means $R = 1:1$, and less means $R = 1:3$. In all conditions, $T = 0.2s$. Results in the other three conditions can be found in **SI** Fig 1. The error bars here are typically too small to see, so any visible differences here are likely highly significant (see the right panel of Fig 4 **A**). The performance numbers are provided in **SI** Table 1.

ImageNet. To confirm that this result is robust to hyperparameter changes in these algorithms, we tested BYOL with different key hyperparameters and found that it still fails in the lower-diversity conditions across all tested configurations (see **SI** Fig 7).

DINO is an interesting model as it is like BYOL with ViT with additional practices like centering and softmaxing, yet its drop in low-diversity condition is much smaller. However, as the centering operation is very similar to contrasting the current teacher embedding to previous embeddings, the result of DINO is actually consistent with the hypothesis that negative sampling is useful.

Unlike the contrastive learning algorithms, MAE is insensitive to the change of the within-batch diversity, as its performance barely changes with respect to the continual learning conditions. As the performance of DINO with the same ViT architecture is significantly influenced by the diversity, this insensitivity of MAE cannot be due to the ViT architecture it uses. Instead, it is likely due to the fact that its loss formulation focusing more on within-image cross-patch relations, while the general contrastive learning loss formulations focus more on the cross-image relations.

Finally, although the life-long benchmark uses one specific source of developmental egocentric video (SamCam), we find that the results above are highly robust to the specific choice of data source, remaining consistent when evaluated on egocentric videos either from other child subjects, or from adults in the Ego4D [20] (MIT License) dataset (see **SI** Fig 3). However, the child developmental dataset more starkly exposes the gaps between the distinct algorithm classes across all data diversity parameter conditions, which underlies our choice to use it as the benchmark.

*Real-Time Learning Results.* We find that the algorithms that fail in the low-diversity conditions also tend to fail in the real-time benchmark even after aggregating their performance across all tested continual learning conditions (Fig 3 **A**, results of separate conditions are in **SI** Fig 2). Interestingly, MAE completely fails to match human performance changes on the real-time learning benchmark (Fig 3 **B**), which is analyzed later. We further hypothesize that the algorithms explicitly leveraging negative samples in the loss formulations also perform well on low-diversity conditions. This is validated by the results of BYOLNeg and DINONeg, as BYOLNeg outperforms BYOL on both real-time and life-long benchmarks and DINONeg outperforms DINO on the life-long benchmark and performs similarly as DINO on the real-time benchmark.

*Tradeoff between real-time flexibility and the life-long stability.* Although higher within-batch diversity generally leads to better life-long benchmark performance, achieving this through lowering the mix ratio implies less learning from the current context, which intuitively could hurt the real-time learning performance. To evaluate this intuition, we systematically test the corresponding performance on both benchmarks using one of the best-performing models on the real-time benchmark (SimCLR-More-MLPs) under the continual learning conditions with more extreme parameter settings. The per-condition learning effect results for these tests are shown in **SI** Fig 8.

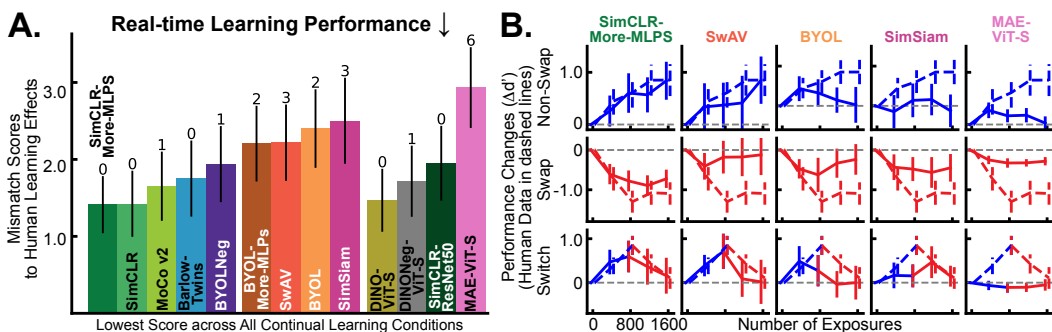

Figure 3: **Real-time learning results. A.** Real-time benchmark performance measured by the mismatch scores to human learning effects. Lower is better and the minimal value is 1.0. Error bars are standard deviations across bootstrapped examples. Numbers above the error bars are the number of datapoints that are significantly different from human data ($\alpha = 0.05$), out of 12 data points. **B.** Learning effects of humans in dashed lines and the unsupervised DNNs under their best conditions in solid lines. In addition to one of the well matching model (SimCLR-More-MLPs), the worse models are shown here. The effects of other models are in **SI** Fig 2. The mismatch numbers are provided in **SI** Table 2.

For the current-memory mix ratio parameter, more learning on the current context (higher ratio) reduces the within-batch diversity, yielding worse life-long learning performance (Fig 4 **A**, left panel). But until $R$ reaches the extremely high value (1:0, meaning only learning from the current context), the real-time learning performance only shows slight increase compared to the base 1:1 value. The significantly larger mismatch when $R = 1 : 0$ is mainly due to catastrophic forgetting, which is further analyzed in **SI** Sec. 1.1.6. When less learning comes from the current context ($R$ lower than 1), the life-long learning performance increases as the within-batch diversity is higher. However, this also means the learning signals needed to match the real-time human learning effects are sparser, which leads to an increase in the mismatch score (Fig 4 **A**, left panel, $R = 1 : 7$ or $1 : 15$). If all learning is from the memory ($R = 0 : 1$), the models are then unresponsive to any changes in the current context, therefore greatly mismatch human learning effects.

Similarly for the current-context replay window parameter, longer replay window increases the within-batch diversity, but also lower the focus on very recent experience. Reflected in the real-time learning benchmark, longer replay window length like 40m or 80m makes the learning slower at the beginning but faster later as well as less human-like in the Switch condition, since the learning signals to the models cannot immediately "switch". Therefore, these longer replay windows lead to worse real-time mismatch scores (Fig 4 **A**, middle panel).

The influence of the aggregation time on the two benchmarks is markedly different. As shown in the right panel of Fig 4 **A**, the life-long learning performance barely changes with respect to $T$, while the mismatch score to human real-time learning effects greatly increases from $T = 0.2s$ to $T = 1.0s$. The change seems even clear for $T = 0.4s$ compared to $T = 0.2s$. The reason for this extremely high sensitivity is that with a longer aggregation window, the chance of sampling the aggregation pairs that represent the wanted learning signal is consequently much reduced.

The tradeoff between real-time and life-long benchmark performance clearly suggests that both benchmarks should be jointly tested to complement each other. Algorithms without negative samples therefore perform even worse compared to other algorithms in this joint testing, as the condition with the lowest real-time mismatch score leads to much lower life-long performance and condition with better life-long performance also typically leads to higher real-time mismatch score (Fig 4 **B**).

*Analysis of learning failures.* To further diagnose the failure of models on the real-time learning benchmark, we construct a purified and conceptually simpler (but unnatural) real-time learning stream by manually selecting aggregation pairs for the models to learn from. Specifically, we subselected pairs of consecutive frames in which there are two *different* images of isolated objects, dropping all pairs of frames which contain the same image in both frames or one blank frame during the exposure phase. In other words, this manually-selected pair stream has been highly de-sparsified to contain precisely the events in which a non-trivial learning signals are expected to be present. Compared to the naturally-emerging aggregation pairs sampled from the actual video stream, the manually selected aggregation pairs make the learning signal denser and also less noisy (see **SI** Fig 5 for examples).

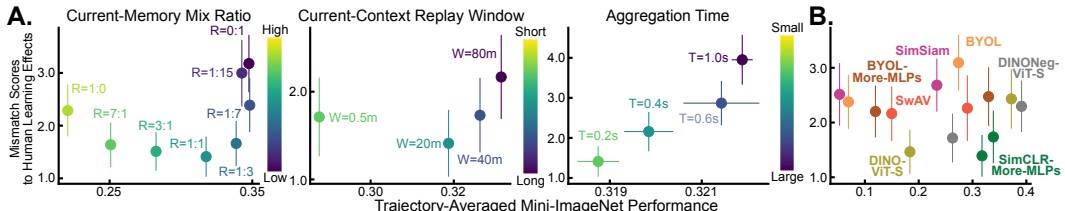

Figure 4: **Tradeoff between the real-time and life-long benchmark performance. A.** Performance of both benchmarks on the SimCLR-More-MLPs model with varying current-memory mix ratio (left panel), current-context replay window (middle panel), and aggregation time (right panel). X-axis represents the life-long benchmark performance. Y-axis represents the real-time benchmark performance. The base setting is $R = 1 : 1, W = 20m$, and $T = 0.2s$. Note that the range of x-axis becomes increasingly small from left to right. X-axis error-bars in the right panel represents the standard error of the means from three models trained with different random seeds. **B.** Performance of both benchmarks for selected algorithms under both the continual learning condition with highest life-long performance and the condition with lowest real-time mismatch.

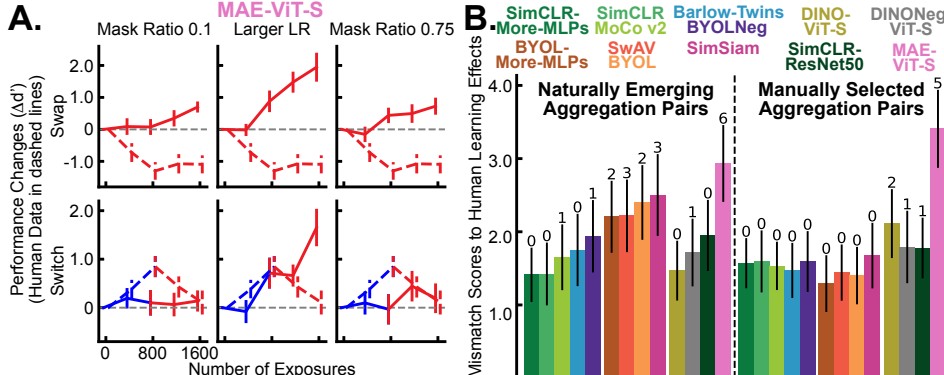

Figure 5: **Analysis of the real-time benchmark results. A.** Learning effects of MAE using manually selected aggregation pairs under different settings. **B.** Mismatch scores to human learning effects for both naturally emerging (left) and manually selected (right) aggregation pairs.

After evaluating all the algorithms in this de-sparsified learning stream under the highest-diversity learning condition tested ($R = 1 : 3$ and $W = 20m$), which has the least focus on the current context, we find that MAE still shows substantial mismatch, and is in fact now the only tested algorithm to do so (Fig 5 **A, B**). Most surprisingly, its learning effects in the Swap condition show *increasing* discriminative performance, unlike all other models as well as human subjects (Fig 5 **A**). We considered the possibility that this was due to the fact that the (default) high mask ratio MAE used (0.75) could be obscuring important details differentiating the two objects. However, even after reducing it to 0.1, MAE still fails to show a decreasing performance in the Swapped condition (Fig 5 **A**, left and middle panels). The slight decreasing performance shown in Fig 3 **B** is possibly due to learning from the pair containing the exposure and gray images. These results suggest that the masked autoregressive loss formulation, with no mechanism to construct semantically meaningful features that are invariant across augmentations, may be at a disadvantage in capturing the flexibility of human real-time learning effects.

In contrast to MAE, almost *all* the contrastive algorithms achieve noise-ceiling level performance on the real-time learning benchmark in the manually de-sparsified learning stream (Fig 5 **B**). This result shows both that these algorithms are capable of capturing the temporal statistics learned by humans as long as key candidate learning events are identified *post-hoc*, and that the failure of those models without negative sampling are specifically due to their inability to automatically identify the learning signals in such events when they arise in the noisier and sparser natural learning stream.

## 5    Discussion

We introduce a real-time human learning benchmark measuring how well unsupervised models predict human visual learning effects and a life-long learning benchmark measuring how efficient

these models learn under a human-generated continual learning curriculum. We further propose a general continual learning process where models jointly learn from the visual experience sampled from a recent time window in the current context as well as memory. Multiple high-performing self-supervised learning models are evaluated and differentiated using the proposed benchmarks. Our results show that the newly proposed algorithms like SwAV, BYOL and MAE underperform earlier proposed algorithms like SimCLR and MoCo v2 on both the real-time human learning and the conditions of the life-long learning benchmarks that lead to lower within-batch diversity, even though these newer algorithms all have been reported to outperform earlier ones on the typical ImageNet dataset. We further show that the algorithm design of explicitly leveraging the negative samples indeed helps the performance on both benchmarks by showing that a variant of BYOL using negative samples performs much better on both real-time and life-long metrics. Through more analysis on the failure of these models, we identify that the failure of some of the learning algorithms is likely due to their inability in learning from the sparse signals from the low-diversity environment.

Our formulation has a number of limitations. Although the current design of the continual learning process uses joint training on memory and the current context to address catastrophic forgetting, another potential solution for this issue is to apply general-purpose continual learning methods such as EWC [37]. However, our preliminary results show that this method is unlikely to improve life-long learning performance even compared to the pure continual learning setting ($R = 1 : 0$) (see **SI** Sec. 1.2.6 and Fig 6). Designing improved learning algorithms that explicitly integrate memory to prevent catastrophic forgetting may thus be helpful. In addition, the current random sampling policy from the short aggregation time window, current-context replay window, and memory can also be improved. Furthermore, humans actively interact with their surrounding environment and effectively choose what they learn from through choosing what they attend to. This feature is not yet captured in our benchmarks, as the real-time learning benchmark evaluates the learning dynamics from the controlled visual stimulus and the life-long learning benchmark presents the models the visual experience that was interactively generated by the children at the time of recording but is fixed for the models. There have been works integrating such interactive curriculum learning into the learning algorithms, especially in exploring how curiosity can help the agents explore or learn in human-like fashion [23, 48, 19]. Enabling the evaluation of such feature in our benchmarks is therefore another important future step.

It is well known that young children undergo a critical period in their visual development [2], suggesting that the underlying learning algorithms or even architecture undergo substantial changes at some point. However, in this work, we do not account for this directly. We simply use the potential changes of learning rates to accommodate such a difference, where smaller learning rates are typically used for the real-time learning benchmark. It is possible that our simple learning-rate schedule *is* a reasonable null model of developmental changes to start with, but testing other more sophisticated models (e.g., fixing lower layers earlier in training) will be part of future work. Moreover, our current benchmarks seek to model only behavioral learning effects [30], but comparing models to learning effects at the scale of individual neurons [43] will be a key future step.

As we train DNNs using standard backpropagation algorithm, it is unlikely that this optimization procedure is implementable in real organisms [4]. Noticing recent progress in local learning rules that are more biologically plausible and the closing gap between these algorithms and error feedback [40], we also plan to combine these new rules with unsupervised learning objective functions to test whether the combined models can explain the human learning effects better in future work.

Finally, the egocentric videos recorded from infants were from middle-class families living in the United States and Australia, making the videos unrepresentative of communities with different socioeconomic statuses or different cultures. Although we believe the conclusions from the life-long learning benchmark will hold for visual experience from children with different background, which is supported by the high consistency between results from SamCam and AliceCam (see **SI** Fig 3.B), collecting recordings from children of more diverse backgrounds will still be an important future step to enhance the inclusiveness of the benchmark. The SAYCam videos also contain personally identifiable information as the faces of the parents and the infants can appear in the videos, which has been consented to by the parents participating in that project [51].

## Acknowledgments and Disclosure of Funding

C.Z. is supported by the MIT ICoN Postdoctoral Fellowship. D.L.K.Y is supported by The Simons Foundation Collaboration for the Global Brain and the NSF CAREER Award. This work is also supported by NIH R01 MH069456 (N.T., K.N., and D.Y.), National Institutes of Health to J.J.D. (2-RO1EY014970-06), Simons Foundation (SCGB [325500], [542965] to J.J.D.), and Office of Naval Research (MURI [N00014-21-1-2801] to J.J.D.). We thank Stanford HAI Institute for their support in cloud computing credits on Google Cloud, AWS, and Azure.

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
