# OpenReview forum: "How Well Do Unsupervised Learning Algorithms Model Human Real-time and Life-long Learning?"
_NeurIPS.cc/2022/Track/Datasets_and_Benchmarks — NeurIPS 2022 Datasets and Benchmarks _

### Official Review · Reviewer_d4cU · 2022-07-26
**Interesting ideas and benchmark, held back by a lack of clarity in writing and too dense structure**

**Rating:** 6
**Confidence:** 3

**Strengths:**

The main strength of the paper is that it tackles an important challenge of human-like lifelong learning from a complementary perspective of short and long-term knowledge acquisition and retention.  It’s ambitious, but there a great promise and many prospects from the type of investigations conducted on the two suggested benchmarks. In addition:

* There introduction and literature review sections are well written and intuitive to read.
* The visualizations, albeit partially a bit too small, are useful and well done.
* The empirical evaluation seems quite comprehensive, with many different types of self-supervised approaches considered, different architectures e.g. a visual transformer, and analysis of trade-offs such as the mask ratio, replay window sizes, and the “memory mix” ratio.


**Weaknesses:**

In principle, I believe the paper to be a great asset for the community. However, it is also a bit tough to vote for acceptance of the paper in its present form, despite its many strengths.

Primarily, this is due to the fact that sections 3 and 4 are very hard to follow. The content is extremely dense, crucial details on the exact experimental set-up, considered methods and evaluation are missing, particularly when inspired form prior works that not everyone will be aware of.
Examples are lines 142, 145, 165, 172, 250, 304, caption of figure 2, the role of pre-training on ImageNet/faces etc. Overall, the content suffers from the style of these two sections, where an attempt has certainly been made to exhaustively mention all aspects, at the cost of them being comprehensive enough. In essence, a reader is required to know the nits and bits of prior works, the way the datasets have been acquired (to understand the construction of the benchmarks), all self-supervised methods and their differences etc. To some degree this is reasonable, but even after rereading the sections multiple times it was hard to follow the train of thought and perceive the overall story. When lookin at the figures I felt lost to search for what precisely the captions mean, e.g. “life-long benchmark performance measured by the trajectory averaged Mini-ImageNet performance”, how precisely window means W= 20m etc are defined mathematically. I acknowledge that it may be there somewhere, but even after looking through the supplementary it was tough to find the necessary details.

In similar spirit, the supplementary material was unfortunately less helpful than it could have been. I was hoping to find more details on the above mentioned aspects, more thorough descriptions of the dataset composition, the descriptions of the investigated methods, and additional results.  Overall the material was rather short however, following the same bullet-point list style structure of the main body that only attributes a few lines per topic. The multiple figures that are simple listed at the end are left unreferenced in text, leaving a lot of the heavy lifting to the reader.  As easy as this seems to fix, it presently imposes a large barrier for me to truly assess the paper’s main contributions.

**Additional Feedback:**

I don’t think this is a concern necessarily, perhaps it is indeed more of a question, so I will list it under additional feedback.

While reading the paper, I was a bit surprised about the choice of mechanism and why it has fallen on a raw experience replay perspective. Surely it is the easiest to implement, but following the many arguments on machine learning models that learn like humans and trying to investigate whether they do etc. I am wondering whether experience replay is at all realistic. Already in the old mid-90s connectionist pseudo-rehearsal papers by Robins or French, the argument has been made that from a neuroscientific perspective replay of original data instances seems rather unlikely, which is why many modern biologically inspired lifelong learning perspectives go with a mix of regulating the stability-plasticity trade-off and employing pseudo-rehearsal mechanisms, see e.g. “biological underpinnings for lifelong learning machines”, nature machine intelligence 4, 2022.  For most of the experimental take-aways it likely does not matter here, but much of the benchmark framework also seems to be built around the concrete interleaving of stored data. I would be curious to have a more detailed statement on the made choice and trade-offs here.

**Clarity:**

As mentioned as the primary point in the weakness section, sections 3 and 4 suffer from a lack of clarity, primarily due to the compressed very dense nature of the writing and a structure that quickly “moves on”.

**Correctness:**

I believe the benchmark construction and investigation to be correct, but simultaneously will emphasize that it is hard to truly make a definite statement here, given the lack of clarity stemming from the very compressed writing.

**Documentation:**

The constructed benchmarks would benefit from a more thorough documentation, even if this means that portions of prior works are simply summarized.

**Ethics:**

I don’t think there’s an ethical concern necessarily. However, it is odd that a benchmark containing recordings from a child’s perspective commits the ethical considerations section and lists them as “not applicable” in the checklist. The idea is to talk about societal and ethical implications, even if the data originates from prior works that did not necessarily provide such a discussion.

**Relation To Prior Work:**

The literature review nicely relates to various works from different fields, providing a good overview of the streams of perspectives across communities.

**Summary And Contributions:**

The paper introduces two sets of benchmarks, one on the real-time learning towards assessment of rapid changes, one on the life-long side in an online long-term curriculum. Here, the inspiration and data foundation is taken from a child’s stream of visual experiences over multiple years. In this context, various self-supervised learning algorithms are investigated, contrasted and analyses with respect to their ability to perform adequately on both benchmarks and mimic the human learner. Most importantly, choices and trade-offs are investigated when the two benchmarks are regarded from a complementary perspective.

Post rebuttal update: see response below, rating raised from 5->6

---

> ### Author Response · Authors · 2022-08-11
> **Response (1/2)**
>
>
> The main concern of the reviewer is the clarity of the method section in our paper, which we have addressed through updating the paper and __SI__.
> More specifically, we have added more details in the method section, together with pseudo-code algorithm descriptions of how the benchmarks are built in __SI__ (see Alg. 1 to 3).
> We have also added more loss definitions of unsupervised methods to __SI__ (see Sec. 1.2.3).
> We hope the updated version can help the reviewer better understand this work.
>
> The reviewer also mentioned the need to include societal and ethical implications, which are now added to the updated paper (see the last paragraph of the Discussion section).
>
> Finally, the reviewer listed an additional feedback about why the choice of mechanism "has fallen on a raw experience replay perspective".
> More specifically, the reviewer mentioned the alternative methods which are "a mix of regulating the stability-plasticity trade-off and employing pseudo-rehearsal mechanisms".
>
> A key thing to note is that nothing in our benchmarks is specifically tied to using replay, and in the Discussion section of this paper, we have mentioned the interesting future possibility of using other continual learning methods that avoid direct experience replay.
> Indeed, in the existing work we have tried one representative such method (Elastic Weight Consolidation) and found that this method barely helps the performance, that is, the DNN trained with EWC achieves no better than the DNN only trained on the current context (see __SI__ Sec. 1.2.6 and Fig 6).
> So far in our tests, replay-like mechanisms are the best performing method we have tested and DNNs with this mechanism not only achieve much better performance on the life-long learning benchmark, but also reaches nearly ceiling score on the real-time learning benchmark.
> At least on the benchmarks built in this work, models with the replay mechanism better explain the human learning dynamics in both real-time and life-long time-scales.
>
> Of course, it's important to note that our benchmarks are constructed in a way that is agnostic to exactly what the memory mechanism is, so a future method that has no replay could in principle do well in our benchmarks, and we would very much welcome that kind of contribution. Moreover, the analysis we presented in our work about how the continual learning process influences the algorithm performance on both benchmarks is also generalizable to other non-replay methods, as the current context replay window $W$ and aggregation time $T$ will also be key parameters there and the mix ratio $R$ can be generalized to the ratio between the "stability" and the "plasticity".
> Ultimately, the reason why we've contributed our work as a benchmark and dataset is to spur algorithm development in multiple directions (including away from explicit replay). We will make this clearer in the revised paper.

---

> ### Author Response · Authors · 2022-08-11
> **Response (2/2)**
>
>
> The reviewer also mentioned that "Already in the old mid-90s connectionist pseudo-rehearsal papers by Robins or French, the argument has been made that from a neuroscientific perspective replay of original data instances seems rather unlikely".
>
> We want to first argue that the previous arguments are far from conclusive and mostly open-ended.
> The evidence presented then was also quite preliminary, and countervailing arguments have been made since [1, 2, 3, 4].  Ultimately, existing works have largely been limited to very small datasets that do not conclusively settle the question one way or another.  This is one of the motivations behind our current, much larger-scale, evaluations.
>
> If strong data that can totally rule out the replay mechanism exist, we are more than happy to include such data into our benchmark and re-rank the algorithms based on that. In fact, it *could* have been the case that replay mechanisms were ruled out by our existing benchmarks, and in that case, we would have reported that result as an interesting benchmarking failure.
>
> However, as the case actually turned out, the human learning dynamics we are modelling is best explained by the models with replay, so we feel that this mechanism cannot be simply ruled out due to previous preliminary and inconclusive discussion. (Of course replay might certainly be ruled out by future data!)
>
> Finally, it is certainly possible that mechanisms that do not use explicit replay of previous instances but some kind of modified memory-based completion reactivation could certainly be valid.  It would be super-interesting if it turned out that such methods could do better overall on our benchmarks than any method we've evaluated in the present paper.  That is why we've decided to contribute our work as a Dataset and Benchmarks paper, because we hope that by making our datasets and evaluation procedures public and accessible, future researchers will provide even better mechanisms that go beyond the limitations of the specific algorithms we've tested here.
>
> [1] Karlsson, M. P., \& Frank, L. M. (2009). Awake replay of remote experiences in the hippocampus. Nature neuroscience, 12(7), 913-918.
>
> [2] Eagleman, S. L., \& Dragoi, V. (2012). Image sequence reactivation in awake V4 networks. Proceedings of the National Academy of Sciences, 109(47), 19450-19455.
>
> [3] Lu, J., Luo, L., Wang, Q., Fang, F., \& Chen, N. (2021). Cue-triggered activity replay in human early visual cortex. Science China Life Sciences, 64(1), 144-151.
>
> [4] Liu, Y., Dolan, R. J., Kurth-Nelson, Z., \& Behrens, T. E. (2019). Human replay spontaneously reorganizes experience. Cell, 178(3), 640-652.

---

> > ### Comment · Reviewer_d4cU · 2022-08-15
> > **Changes in the revision improve clarity**
> >
> > I thank the authors for their detailed response and the uploaded revision.
> >
> > In particular the supplementary material extension is helpful and already improves the clarity aspect of the paper.
> >
> > As a consequence I am raising my score to now tend towards acceptance of the paper.
> > As the main body of the paper is still rather dense, also in terms of the figures containing a lot of information with long captions that still would require more details, I encourage the authors to further tune this aspect of their work.
> >
> > In addition, I appreciate the discussion and references posted in the second part of the response. It would be great to see this more general discussion also further reflected in the paper.

---

### Official Review · Reviewer_MVr7 · 2022-07-27
**Interesting paper that needs some clarifications and a better written methods section**

**Rating:** 7
**Confidence:** 3

**Strengths:**

**Originality**: Training recent unsupervised methods on approximations of a human curriculum and comparing them is interesting and valuable. The paper draws on evidence from real human learning experiments.

**Quality**: The authors compare a number of current unsupervised learning algorithms and give error bars for them. The claims about negative sampling are well supported. The authors appear to be careful and honest in evaluating the strengths and weaknesses of their work.

**Significance**: The results of negative sampling are "nice to know." The trade-off is interesting. The benchmark itself is important.


**Accessibility and accountability** The authors provide their implementation, which benefits accessibility.

**Relevance to the broader research community**: The benchmarks appear to be relevant to communities engaged in human-like learning. The authors' results are also interesting for the field of unsupervised and continuous learning.


**Weaknesses:**

**Originality**: A comparison with the literature on human-like learning curricula in deep learning seems too sparse, so it is not clear how exactly this work curricula differs from previous contributions in this area.

**Quality**: The claim that a human-like algorithm has to struggle to achieve the discussed trade-off between the two benchmarks (line 23-27) seems far-fetched. This tradeoff is discussed in terms of batch diversity, but this would require that human-like algorithms learn batch-style. This claim may be true for current unsupervised methods, but does it hold for true human-like learning?

**Relevance to the broader research community**: As explained in more detail in the "Clarity" section, the benchmark (especially the real-time learning benchmark) is not described clearly enough. In addition, the results are only presented in graphs and not in simple numbers (in a table), which makes it difficult to compare them with the given baselines for further work. At least the supplementary material should contain plain numbers. Also, the baseline weights could be shared with the research community.

**Ethical and social implications**: Ethical and social implications are not discussed. The authors believe that there are no potential negative social implications. This statement seems too harsh.


**Additional Feedback:**

Q1: “The gray image serve as a proxy for the visual inputs of human subjects during inter-trial intervals.” Why are the images grayscale?

Q2: You state that “Models learn from the whole data stream including both test and exposure phases, each of which takes 10 minutes” Why does the exposure face exactly take 10 minutes? Does the number of images the model sees then depend on the model's prediction speed?


**Needs (spell) checking (reviewer is not a native speaker)**:
- Appendix: More MLPs &#8594; Larger MLPs
- Did you discuss whether the data you are using / curating contains personally identifiable information or offensive content? We do not think the data we are using contain this. &#8594; the datasets used in the real-time learning benchmark contain human faces

**Limitations**: The authors addressed the limitations of their work, which clarified questions that arose while reading the paper.

**Reviewer Statement**: The reviewer is willing to increase the score if the concerns are addressed by the authors

**Based on the revision 1 and 2, the reviewer increased the score to 7 (see the Reviewer Response to the Revision comment and the Reviewer Response to Revision 2 comment for more details).**


**Clarity:**

The reviewer found the methods section of the paper difficult to read. A different outline could alleviate this problem. For example, Figure 1 could be divided into its subfigures. For each benchmark (Real-time Learning Benchmark, ...), one of these sub-figures could be used to describe the benchmarks, after which the experimental results should be presented. Also, the description of each benchmark should be more consistent with the figures, and the figures should be more consistent with the training process (e.g., a literal mouse click is required in the figure. E.g., the ms in the figure and text do not always match. Figure 1A, for example, is basically the same as in the paper from which it was taken).  In addition, the description of each benchmark could be supplemented with a more detailed algorithmic description (e.g., pseudocode) to facilitate understanding of the benchmarks.

**Correctness:**

To the best of the reviewer's knowledge, most but not all of the assertions in the paper are correct. As mentioned earlier, the trade-off assertion may not apply to a real human-like algorithm. However, the reviewer did not attempt to replicate the results of the experiment part. The evaluation methods seem to be correctly designed, but the lack of plain numbers is a problem and affects the comparison with the paper.

**Documentation:**

The implementation is provided by the authors together with the data loaders and the training program. This supports reproducibility. To further facilitate reproducibility, the results should be provided in plain numbers and the model checkpoints could be provided by the authors.

**Ethics:**

Ethical and social implications are not discussed. The authors believe that there are no potential negative social implications. This statement seems too harsh.

**Relation To Prior Work:**

The relationship to previous work in unsupervised learning, real-time and continuous visual learning in real organisms, unsupervised deep neural network models for the visual system, curricula, and lifelong learning for neural networks is clear. Comparison with the literature on human-like learning curricula in deep learning appears sparse.

**Summary And Contributions:**

The paper presents two benchmarks for human-like real-time and lifelong learning. The real-time learning benchmarks focus on comparing the visual classification behavior of unsupervised learning algorithms with that of humans captured in different sessions. Lifelong learning compares the continuous learning behavior of various unsupervised learning methods by training them on first-person videos of young children using a "child-like" learning curriculum. Different unsupervised methods (BYOL, SimSiam, SwAV, MAE, SimCLR, MoCov2) are evaluated using this benchmark. It is shown that negative sampling is beneficial in these low diversity benchmarks. In addition, a tradeoff that unsupervised methods must make in order to perform well in both benchmarks is discussed.

---

> ### Author Response · Authors · 2022-08-11
> **Response for the Weakness comments (1/2)**
>
>
> - *"Originality: A comparison with the literature on human-like learning curricula in deep learning seems too sparse, so it is not clear how exactly this work curricula differs from previous contributions in this area."*:
>
>     To the best of our knowledge, the real-time learning benchmark in our work is the first effort to use stimulus-based computational models to explain the human data collected by Jia et al. In fact, we are not really aware of any comparable comparison of data about real-time human visual learning (at e.g. the minutes or hours time scale) with computational models that explain such patterns in terms of the actual visual statistics of the stimuli to which the human is exposed.  However, we would be very happy to have a look at and relate our work to any papers that the reviewer could suggest (that would be very helpful!).
>
>     As for the life-long learning benchmark, the SAYCam is also a very new dataset released only within the past several years. Although there are previous works using this dataset to train DNNs (refs [36, 61] in the paper), they used this dataset in an offline fashion, which is very different from how we use this dataset.
>
> - *"The claim that a human-like algorithm has to struggle to achieve the discussed trade-off between the two benchmarks (line 23-27) seems far-fetched. This tradeoff is discussed in terms of batch diversity, but this would require that human-like algorithms learn batch-style. This claim may be true for current unsupervised methods, but does it hold for true human-like learning?"*:
>
>     The nature of human learning is not at all clear.  It might indeed be batch like, or it might not.
>
>     The case for its being batch-like is not without some evidence. Just from a basic psychophysical and retinal sensor point of view, it's unlikely that humans literally update totally independently on each frame of input. Also there is some evidence (both from the neurophysiology and psychophysical literature) for temporal normalization and adaptation which could be consistent with some kind of batch statistics being relevant to learning at both the short and longer-term scales.
>
>     On the other hand, there is no strong evidence that human learning is exactly batch-like either.  Previous discussions in the literature about this point are really inconclusive and mostly open-ended for both answers, especially as the batches in our benchmark can include both the stimulus from recent visual experience and the previous memory.
>     For example, studies have shown that activity or experience replay can happen in early or later visual cortex [1, 2], which effectively influences the learning at the current moment like how batched learning would.
>
>     This uncertainty is exactly the point of an effort like ours. Rather than trying to use a very inconclusive literature to build in some assumption of what human learning must be like, we instead have here sought to build a benchmark that could allow algorithms (with or without batch-like mechanisms) to be evaluated for their ability to match the data.   There is nothing in our benchmarks themselves that require batch-like computation as such.   Thus, in principle, an algorithm without any batch structure could do very well on our tests, and if it did, that would be quite interesting.
>
>     Now, it is true that in evaluating specific algorithms against our benchmarks, the algorithms do use batch-like updates. This is essentially because it seems in practice extremely difficult to find algorithms that do any meaningful visual learning at scale that do not have some form of batching.  This observation is by itself a (weak but meaningful) suggestion that human learning might indeed use some batching. Moreover, some of the current unsupervised learning methods almost fully explain the real-time learning dynamics of humans, which is a stronger form of such evidence that batching might be responsible for real human learning dynamics.   Obviously this is not conclusive, but given the *a priori* uncertainty around what human-like or not, the results of our benchmark tasks shed meaningful light on the topic and are a good jumping-off point for further work seeking to determine the role of batching in human learning.
>
>     Ultimately, however, the way that our benchmarks are constructed is itself a useful diagnostic as to whether learning is in batches or not, and we very much welcome future methods without using batches to be evaluated on our benchmarks!
>
>     [1] Eagleman, S. L., \& Dragoi, V. (2012). Image sequence reactivation in awake V4 networks. Proceedings of the National Academy of Sciences, 109(47), 19450-19455.
>
>     [2] Lu, J., Luo, L., Wang, Q., Fang, F., \& Chen, N. (2021). Cue-triggered activity replay in human early visual cortex. Science China Life Sciences, 64(1), 144-151.

---

> ### Author Response · Authors · 2022-08-11
> **Response for the Weakness comments (2/2)**
>
>
> - *"Relevance to the broader research community: As explained in more detail in the "Clarity" section, the benchmark (especially the real-time learning benchmark) is not described clearly enough. ..."*:
>
>     The paper and __SI__ have been updated to address the clarity concern about the method section. The reviewer suggested splitting the first figure by the benchmarks. We appreciate this suggestion, but we believe putting the framework of the two benchmarks together helps illustrate the shared learning mechanisms between them and creates a clear view on how the key parameters (like the aggregation time) are consistently instantiated on both benchmarks. The reviewer also mentioned the inconsistency between Fig 1A and the text descriptions, which we have clarified through adding the note that Fig 1A is the testing pipeline for human subjects in the caption. We present the schema for humans as we think this should help understand the nature of the human data we are comparing to. The testing pipeline for the models is in principle very similar to this and has been described in more details in the texts in the method section as well as figures and algorithms in __SI__.  We have also provided tables listing actual performance numbers in __SI__ Table 1 and 2. As for the baseline weights, they were in fact provided in our Supplementary Materials, which can be downloaded by following the instructions in the source codes. We also thank the reviewer for the suggestion of supplementing the description with "a more detailed algorithmic description (e.g., pseudocode)", which is now added to the __SI__ (see Alg. 1, Alg. 2, and Alg. 3).
>
> - *"Ethical and social implications are not discussed. The authors believe that there are no potential negative social implications. This statement seems too harsh."*:
>
>     We have updated our paper to include discussions on this, please see the last paragraph of the Discussion section.

---

> ### Author Response · Authors · 2022-08-11
> **Response to the additional feedback**
>
>
> - *"Why are the images grayscale?"*:
>
>     The experiments on human subjects performed by Jia et al. used grayscale images as the inputs. So we followed their paradigm.
>     The gray background images used between trials are also approximations for the inter-trial stimulus human subjects perceive during experiments.
>
> - *"Why does the exposure face exactly take 10 minutes? Does the number of images the model sees then depend on the model's prediction speed?"*:
>
>     This 10-min duration of the exposure phase is a close approximation of that for human subjects. Although we conveniently set this as a constant in our benchmark design, we have confirmed that reasonably varying this number does not change our key conclusions. The number of exposure trials is always 400 in one exposure phase for both humans and models.
>
> - *"Appendix: More MLPs → Larger MLPs"*:
>
>     Thanks for this suggestion. The models with More-MLPs are variants of the original models but with more layers in their MLPs, so we believe "More-MLPs" is more indicative of the actual modification than "Larger-MLPs".
>
> - *"Did you discuss whether the data you are using / curating contains personally identifiable information or offensive content? We do not think the data we are using contain this. → the datasets used in the real-time learning benchmark contain human faces"*:
>
>     We have updated our paper to include discussions on this. However, we do not think the data used in the real-time learning benchmark contain personal identifiable information as the human faces used there are from 3D models of human faces instead of real humans.

---

> ### Comment · Reviewer_MVr7 · 2022-08-12
> **Reviewer Response to the Revision**
>
> The reviewer wants to thank the authors for their detailed feedback, for incorporating recommendations into their paper, and for answering Q1 and Q2.
>
> Based on the revision, the reviewer raised the score to 6.
>
>
> *Strengths of the revision*:
>
> **Clarity**: The authors improved clarity by rewriting the method section and adding pseudocode.
>
> **Documentation**: The authors added plain numbers of the results to the supplementary material, which eases the comparison with the proposed work. The model checkpoints were already available.
>
> **Ethics**: The authors added an appropriate ethical discussion and highlighted that the images from Jia et al. were generated.
>
> *Weaknesses of the revision*:
>
> **Correctness**:  The claim that a human-like algorithm has to struggle with the discussed trade-off (lines 23-27) still seems far-fetched. The reviewer thinks another answer from the authors is needed to resolve all doubts.
>
> The author’s argumentation, as far as the reviewer understands, is as follows:
>
> it is unclear how humans learn &rarr;  the paper collects evidence for the trade-off of current unsupervised methods &rarr; since these unsupervised methods are trained with a human-like curriculum, human-like learning algorithms must suffer from this trade-off too.
>
> However, since it is still unclear how humans learn, there seems to be no guarantee that humans suffer from this trade-off too. Isn’t it possible that humans learn without this trade-off or that, in theory, one could build a human-like algorithm without this trade-off?
>
> As far as the reviewer understands, the paper collects evidence that such a trade-off could also exist in human learning but to declare that it **must exist** requires further evidence.
>
> **Clarity**: Although the methods section has improved, the reviewer feels that Figure 1 could still be improved. The figure still appears too cluttered. A division of the figure or a clearer layout/design should be considered.
>
> **Relation To Prior Work**: Comparison with the literature on human-like learning curricula in deep learning still appears a little sparse. The reviewer has no doubts about the originality of the work, but maybe other works that utilized “human-like” learning curricula can be discussed in the context of this paper. One example could be “Goal-and-Curiosity driven Curriculum Learning” [1]. The reviewer is not an expert in human-like learning curricula but believes that a discussion with such related work can help to understand which “human-like” learning curricula exist in the deep learning literature. Furthermore, the reviewer is well aware that the learning curriculum from [1] is used in RL and not in unsupervised learning.
>
> [1] FANG, Meng, et al. Curriculum-guided hindsight experience replay. Advances in neural information processing systems, 2019, 32. Jg.

---

> > ### Author Response · Authors · 2022-08-15
> > **Response (1/2)**
> >
> >
> > We want to first thank the reviewer for quickly responding to our response, increasing the score, and continuing to give us comments.
> > We address these comments as follows:
> >
> > - *"The claim that a human-like algorithm has to struggle with the discussed trade-off (lines 23-27) still seems far-fetched. The reviewer thinks another answer from the authors is needed to resolve all doubts."*:
> >
> > Indeed, we agree with the reviewer that our original claim, "... induces a trade-off that human-like algorithms must straddle", is stronger than what we really are licensed to make (or even meant to make, honestly).
> > We have thus modified that claim to be "... induces a trade-off that human-like algorithms *may have to* straddle".
> >
> > But expanding on this idea a bit, we feel that the chain of inferences that the reviewer described is not exactly what we have in our mind.
> > So we want to use this opportunity to clearly explain the logic chain in our mind.
> > Essentially, what we’re doing is (a) starting with a phenomenon (the underlying nature of human learning) whose nature is unknown because it is extremely difficult or impossible to measure directly (b) setting up a measure of correctness at matching that phenomenon (c) asking if a given model matches it (d) looking at implications of some of the models that seem to match it to some extent.
> > Obviously we have no guarantee that the implications of (d) necessarily hold. But as we find that the measures of correctness in (c) improve, we get stronger and stronger inferences.
> > We definitely don’t want to be seen as saying that the tradeoff must exist in humans --— that’s definitely a stronger statement than we want to make.
> > But we find that (a) across as wide a class of algorithms as we can implement right now the tradeoff exists and (b) we have set up an independent  measure of whether the models are correct.
> > We (and we hope others) will with increasing accuracy be able to infer whether the tradeoff exists in humans by improving on the measures of correctness over time.
> > The reviewer is totally right that more data is needed to assess whether the tradeoff is likely to exist in humans.  We think of our work as a good framework for processing and organizing the efforts toward generating that further data.
> >
> > Thus, we'd like to propose a modification of the reviewer’s “chain of inference” to be as follows:
> >
> > "It is unclear how humans learn → the paper leverages data about the empirical pattern of human learning to design benchmarks of correctness → the unsupervised methods can be measured for correctness in matching human data when trained the human-like curricula → to the extent that the models match empirically-collected human-like learning patterns the true human algorithm is more likely to suffer from this trade-off too → if a better algorithm class comes along that matches the empirical data better but doesn't suffer from the tradeoff, that reduces our likelihood for thinking the tradeoff would apply → over time we hope the chain of inferences converges on a most-likely answer."
> >
> > One of our contributions is to have identified datasets and created metrics and benchmarks that will underlie progress on this chain of inference.
> >
> > A second type of contribution is to show that existing AI algorithms are substantially below human levels — and *are* subject to the tradeoff — and this serves as a good target for algorithmic improvement regardless of whether the true human algorithm class also suffers from the tradeoff to some extent.
> >
> > In summary, we aren't really trying to make a strong comment in this paper as to whether the human experiences the tradeoff.  Rather, we have found that models of the class that currently come anywhere close to matching the data do experience the tradeoff. But we have also, equally importantly, found that none of the algorithms matches the data perfectly, so none of the algorithms we have found is a great approximation of the true human algorithm yet. Thus, when a better computational approximation for the true human algorithm is found, there are two possibilities: (a) it will be like the current algorithm class in terms of basic assumptions that lead to the tradeoff, but just better in other ways we don't yet understand or (b) it will be better precisely because it violates some of the assumptions of the current algorithms and in doing so escapes the tradeoff. We think either alternative is equally probably, and finding (b) would be very exciting.  We hope that in the future our work spurs people to look into that possibility.
> > Finally, we really appreciate the opportunity to have this important discussion about the nature of data-driven inference at the intersection of AI and cognitive/neuroscience and we are happy to modify the paper to reflect what we've learned here.

---

> > ### Author Response · Authors · 2022-08-15
> > **Response (2/2)**
> >
> >
> > - *"... the reviewer feels that Figure 1 could still be improved. The figure still appears too cluttered. A division of the figure or a clearer layout/design should be considered."*:
> >
> > We have updated Fig 1 to make it cleaner. Thanks for this suggestion.
> >
> > - *"Comparison with the literature on human-like learning curricula in deep learning still appears a little sparse. "*:
> >
> > We thank the reviewer for mentioning this group of works, which in our view explores how human-like interaction with the surrounding environment helps learning. This interaction is indeed an important feature determining human learning, but not yet captured in our benchmarks. We have added sentences addressing this point in the Discussion section, which are copied below:
> > "Furthermore, humans actively interact with their surrounding environment and effectively choose what they learn from through choosing what they attend to.
> > This feature is not yet captured in our benchmarks, as the real-time learning benchmark evaluates the learning dynamics from the controlled visual stimulus and the life-long learning benchmark presents the models the visual experience that was interactively generated by the children at the time of recording but is fixed for the models.
> > There have been works integrating such interactive curriculum learning into the learning algorithms, especially in exploring how curiosity can help the agents explore or learn in human-like fashion [1, 2, 3].
> > Enabling the evaluation of such feature in our benchmarks is therefore another important future step."
> >
> > [1] Pathak, D., Agrawal, P., Efros, A. A., \& Darrell, T. (2017, July). Curiosity-driven exploration by self-supervised prediction. In International conference on machine learning (pp. 2778-2787). PMLR.
> >
> > [2] Haber, N., Mrowca, D., Wang, S., Fei-Fei, L. F., \& Yamins, D. L. (2018). Learning to play with intrinsically-motivated, self-aware agents. Advances in neural information processing systems, 31.
> >
> > [3] Fang, M., Zhou, T., Du, Y., Han, L., \& Zhang, Z. (2019). Curriculum-guided hindsight experience replay. Advances in neural information processing systems, 32.

---

> > ### Author Response · Authors · 2022-08-27
> > **Response**
> >
> > Dear Reviewer MVr7,
> >
> > Please let us know if our followup comments have addressed your concerns about the revision. We would be happy to continue to discuss. Looking forward to your response!
> >
> > Best,
> > Authors

---

> > > ### Comment · Reviewer_MVr7 · 2022-08-29
> > > **Reviewer Response to Revision 2**
> > >
> > > The reviewer thanks the authors for incorporating the suggested feedback into the paper.
> > >
> > > Regarding the discussion, the updated chain of inference clarifies the thought process of the authors, and it feels appropriate to change the *must* statement to *may*.
> > >
> > > Therefore, the reviewer raised the score to 7.
> > >
> > > However, to provide further feedback after rereading the paper, the reviewer encourages the authors to further improve the methods section, especially the algorithmic side of the Real Time Learning Benchmark explanation in collaboration with Figure 1. The "jump" from the human-centered explanation to the algorithmic training side still requires "a lot of effort" to understand. This is a very important section of the paper, so the whole paper benefits greatly from any small improvement in clarity.

---

### Official Review · Reviewer_SJaH · 2022-07-28
**Benchmarks that lead to new insights can benefit the research community**

**Rating:** 7
**Confidence:** 2
**Clarity:** The paper felt a bit difficult to rea…

**Strengths:**

- Rigorous evaluation of models can be messy and non-straightforward. Additional perspectives to evaluate models are good contributions since our DNN models could have other characteristics that conventional evaluation benchmarks cannot adequately capture. The paper presented two benchmarks for evaluating models and for comparing them with how humans learn.
- In addition to the benchmarks proposal, the insights obtained from using these benchmarks were also presented.


**Weaknesses:**

- I am unsure if the paper felt a little challenging to read because of the multiple research areas covered in the paper, some grammatical/writing style choices, or due to the general structure. I cannot spot anything glaring to suggest to make the paper clearer. Hopefully, other reviewers can help in that direction.

**Additional Feedback:**

None

**Correctness:**

The claims are backed by experimental results. Likewise, some of the findings are similar to those I arrived at on a paper currently under elsewhere. One of such is that models such as the MoCo V2/V3 show even more superior performance when subjected to a more rigorous evaluation protocol, which a simple evaluation protocol might not adequately capture.

**Documentation:**

Not applicable.

**Relation To Prior Work:**

I believe the work clearly mentioned the other related works in literature and also introduced some literature that they based their work on.

**Summary And Contributions:**

The paper draws a comparison between how humans and unsupervised learning algorithms learn. Two benchmarks were proposed: one is based on real-time learning, while the other is on life-long learning. Based on these metrics and other analyses, the paper arrived at the following findings:
- newer generations of self-supervised methods show poorer performance based on the proposed metrics, while some older algorithms perform better.
- best performing models have a commonality -- negative sampling
- a major mechanism underlying poor performance based on the proposed real-time is an algorithm's inability to capture the sparse learning signals in low-diversity environments.

---

> ### Author Response · Authors · 2022-08-11
> **Response**
>
> We apologize for the confusion in our method description.
> We have updated our paper and __SI__ to include more details about the methods, as well as including pseudo-code algorithm descriptions about how the benchmarks are built. We hope this helps resolve the confusion.

---

### Official Review · Reviewer_78Zc · 2022-07-28
**Good contributions towards exploring the difference and gap between computer vision models and human vision and learning systems**

**Rating:** 7
**Confidence:** 4
**Correctness:** Yes, it is constructed in a sound way.
**Clarity:** Yes, it is well written.

**Strengths:**

1. The real-time learning and life-long learning tasks are complementary and most related to the gap between AI and human learning attributes.
2. The usage of children's ego video as learning curriculum is technically sound.
3. Several self-supervised learning methods across different styles are evaluated and good insights are provided
4. Discussion on the tradeoff is reasonable and inspiring.

**Weaknesses:**

1. The section of methods of two benchmarks are kind of confusing and cannot be understood easily.
2. It can be more intuitive to present some failed cases of computer vision models and corresponding human results.

**Additional Feedback:**

Line 145 "each trail" may be "each trial".

**Documentation:**

Yes.

**Ethics:**

No.

**Relation To Prior Work:**

Yes.

**Summary And Contributions:**

This paper explores the gap between computer vision models and human vision and learning systems on real-time learning and life-long learning. Several self-supervised methods are evaluated and the results are presented. The reasons behind the difference of different methods are discussed. Following is a discussion on the tradeoff of real-time learning and life-long learning.

---

> ### Author Response · Authors · 2022-08-11
> **Response**
>
> We apologize for the confusion in our method description. We have updated our paper and __SI__ to include more details about the methods, as well as including pseudo-code algorithm descriptions about how the benchmarks are built. We hope this helps resolve the confusion.
>
> As for "some failed cases of computer vision models and corresponding human results", if the reviewer means the failed cases in the real-time and life-long learning benchmarks, we have provided how some learning configurations or some algorithms lead to failures in our paper and __SI__ (for example, see Fig 4 and Fig 5 in the paper and __SI__ Fig 1, Fig 2, Fig 9). If the reviewer refers to other visual tasks that models are still underperforming humans, such as one-shot or few-shot learning, we think it is beyond the scope of this work and would like to address them in future work. We have fixed the typo mentioned by the reviewer, thanks for pointing this out!

---

### Official Review · Reviewer_jYPN · 2022-07-28
**2 benchmarks: Real-time and Life-long learning**

**Rating:** 5
**Confidence:** 4

**Strengths:**

 - Explained the trade-off relationship between real-time and life-long learning. In continual learning or lifelong learning we talk about plasticity-stability dilemma in the scope of a model, but this paper re-interpreted in a more macroscopic perspective by separating it into two benchmarks
 - Showed that the reason current unsupervised models fail on both benchmarks is that they don't utilize contrasting examples

**Weaknesses:**

* The way each benchmark was built is not new. Real-time benchmark used the same method presented in the previous study, but applied online streaming of data instead of offline learning. Life-long learning benchmark is meaningful in that it used children's egocentric video dataset which is differentiated from widely used benchmarks in life-long learning task, but the strategy(choose one child's data, group it into 100 segments, and evaluate every 10 segments) to use it cannot be seen as novel or unique approach.

* Showed that contrastive learning(negative sampling) is important, but few insight or analysis on why contrastive learning works so well when modeling human learning

**Additional Feedback:**

* If SAYCam shoots only for 2 hours every 2 weeks, we might miss many important events that happened during the weeks. I wonder if the two-hour long video can properly describe or support evidence for a child's development of visual recognition ability during the couple of weeks. I assume the authors understand this better, so I would like to hear their opinions.
* Authors might have been working on this topic for a long time, but it would be better to give a more basic explanation that would help those who are entering into this topic or those with a moderate level of knowledge.
* The overall structure can be better organized especially for the methods part. It would be nice if the concepts are explained when they appear for the first time and formulated to clarify the meaning.
* It would be better if the results(numbers) are organized in a table.


**Clarity:**

* The authors passed too many details for the real-time benchmark to the prior work(Jia et al). It is hard to fully understand the setting without reading it. In order for this paper itself to be a good benchmark, the necessary content must be described in this paper. For example, in a real-time benchmark, authors computed the categorization performance measured using d’(line 150), but they didn’t explain the definition of d’ and how to calculate it. Accordingly the next part, “to compute the learning effects through subtracting the changes of d’ on the exposed objects by the changes of d’ on the non-exposed objects” cannot be fully understood. Besides that, it would have been better if the authors explained that the exposure phase is the phase simulating unsupervised visual experience, and the reason for setting 3 experiment conditions (Swapped, Non-Swapped, and Switch).

* This paper focuses on using unsupervised algorithms to model human learning. I think it can be tested only with the real-time benchmark, so it seems unclear why the authors suggested the life-long learning benchmark. I agree with the trade-off between life-long and real-time benchmark, but real-time benchmark already is in online learning and operated with continual learning setting, so it seems life-long benchmark is intervening in the main content that the paper is trying to say. Is benchmarking life-long learning separately a prerequisite for modeling the human brain using an unsupervised algorithm?

* Imitating the way humans learn doesn't necessarily increase real-time and life-long learning performance of deep neural networks, then why is it important for DNN to achieve neural predictivity?

* What is the reason for choosing Sam among 3 children in SAYCam?

* line 161: Why using 4 test phases out of 5?


**Correctness:**

* line 51 : it seems a leap to say that 'unsupervised algorithms can describe the learning dynamics of human behaviors under all time-scales', from the claim that 'they leverage the unlabelled stimuli'. This sentence is at a very important place bridging unsupervised learning to human learning, which is the key hypothesis of this paper, so it should be explained more logically.

* line 162: The three conditions (Swapped, Non-Swapped, and Switch) model different learning environments and the amount of performance increase and decrease is different for each condition, so it would be meaningful to look at them separately, not averaging them. Is there a reason for averaging the results of all conditions?

* line 177: If we want to see whether we can model human learning in temporal fashion through a life-long benchmark, shouldn't we see how the 10 evaluations of model performance change variates from human performance change? Here the authors averaged the results of 10 evaluations even though they contained temporally different evaluation results.

**Documentation:**

There is sufficient detail to support reproducibility.

**Ethics:**

No.

**Relation To Prior Work:**

Stated above.

**Summary And Contributions:**

Humans can rapidly learn from current experience as well as utilize information accumulated over longer periods. This paper tried to suggest 2 benchmarks describing each case and analyze the trade-off between them by evaluating several unsupervised learning algorithms on the benchmarks. Main contribution of this work is to find out that the negative sampling is key to building human-like learning models.

After the revision of the paper, the reviewer raised the score to 5.

---

> ### Author Response · Authors · 2022-08-11
> **Response to the Weakness comments**
>
>
> - *"The way each benchmark was built is not new. Real-time benchmark used the same method presented in the previous study, but applied online streaming of data instead of offline learning."*:
>
>     We're not really sure which previous study the reviewer is referring to here.  Do you mean the original experimental paper introducing the human behavioral data itself [ref 28 in the paper]?
>     That paper is just a report of an experimental phenomena in humans and a comparison of it at a high level to previous neural studies.  However, that paper doesn't make any comparisons of the data to computational models at all (with either online or offline learning), and it (perhaps more importantly) does not introduce a method or metrics for comparing such data to image-computable models (such as DNNs).  What we contribute here is (a) the paradigm for making fine-grained comparisons of human learning trajectory data to computational models, (b) the public release of the data and the benchmarking metrics code, and (c) the actual benchmarking of a series of competitive computational models of the data as a baseline for future work using the benchmark.
>
>     Or is the reviewer referring to other papers in the literature?  Ultimately key contribution of the real-time benchmark is to compare detailed  dynamics of DNN learning to that in humans. To the best of our knowledge, our work is the first one to do so.  We'd love to know about other papers if they're relevant to this issue, so it if this what the reviewer meant, we'd kindly ask you to provide specific references we can respond to.  (Very happy to engage in a discussion about it!)
>
> - *"Life-long learning benchmark is meaningful in that it used children's egocentric video dataset which is differentiated from widely used benchmarks in life-long learning task, but the strategy(choose one child's data, group it into 100 segments, and evaluate every 10 segments) to use it cannot be seen as novel or unique approach."*:
>
>     Hm... we think it's fair to say that our procedure for construction of the life-long learning benchmark involves a lot more than just grouping the child's data into 100 segments! Once segmented, the videos belonging to one segment are used to construct the visual stimuli stream through sequentially presenting the frames within them.
>     This stream is then used to construct batches of inputs to the DNNs using our proposed continual learning process.  This continual learning process involves (i) sampling from a current-context replay window, (ii) integrating stimulus within a short aggregation time, and then (iii) mixing the samples from the current context and the memory.
>     This is a highly non-trivial continual learning process and seeks to naturally capture many aspects of psychology studies on learning -- but to our knowledge has not been proposed in previous technical works comparing real life-long learning data with computational models.
>     Moreover, both the real-time and life-long benchmarks share this process as the underlying mechanism to construct batches from the stimuli stream.
>     This shared design between the two complementary benchmarks is both novel and a critical choice for the discovery of the tradeoff between these two benchmarks. We're not aware of any similar proposal or implementation of such an idea in previous works.
>
>     So it seems a bit confusing to us to describe this as not novel.  Of course, the reviewer might have some better or even more novel benchmarking construction procedure in mind, and we'd be happy to chat about it!
>
>     (Of course, the initial step of just segmenting the videos -- before applying the further steps described above in constructing the datastream -- we agree that that specific initial part of the process is very simple. But shouldn't it be? The concept of temporally sequential sequences is itself fairly straightforward, so in defining the initial segmentations, it might not be necessary or even desirable to innovate.)

---

> ### Author Response · Authors · 2022-08-11
> **Response to the Weakness comments (continued)**
>
>
> - *"Showed that contrastive learning(negative sampling) is important, but few insight or analysis on why contrastive learning works so well when modeling human learning"*:
>
>     Hm, this is a little confusing to us.  We felt that we had provided at least some important insight into this reason.  Looking at Fig 5.B of the paper and the related part of Section 4, we have shown that a de-sparsified (but unnatural) configuration of the real-time learning benchmark enables the lower mismatch of the unsupervised learning methods without negative sampling.
>     This result in turn shows that the contrastive learning methods achieve the low mismatch as they are able to leverage the sparse learning signal in the real-world learning environments, which is also the challenge humans need to solve.
>
>     We feel we have thus presented at least one strong possible explanation on why these algorithms work well when modeling human learning. Of course, there might be future additional analysis and insights that the community could have.
>     Ultimately, our work aims to build strong, publicly available, benchmark(s) that could support the community in exploring these issuesand we're hopeful others will follow up on these ideas as future work.

---

> ### Author Response · Authors · 2022-08-11
> **Response to the Correctness comments (1/3)**
>
>
> - *"line 51 : it seems a leap to say that 'unsupervised algorithms can describe the learning dynamics of human behaviors under all time-scales', from the claim that 'they leverage the unlabelled stimuli'"*:
>
>     This is a good point! But we believe that the reviewer may be making something of a "strawman" caricature of our original statement.
>     Our original sentence was "it is *plausible* that they *might* describe the learning dynamics of human behaviors under all time-scales" (emphasis added for clarity).  This is a bit different from the the much more definitive (and unsubstantiated) claim the reviewer is concerned about. We chose our original wording carefully for this reason.  If the reviewer has an improved phrasing though for more precisely phrasing the thought, we'd be very happy to use it!
>
>     It's also important to understand the contrast we were hoping to make with our statement.  Our claim stands in contrast to the previous state-of-the-art models of visual cortex, which were are supervised models whose training required large amounts of labelled annotations. Since such a vast quantity of annotations are not available in real child learning environments, it is usually considered highly implausible for these supervised models to describe the learning dynamics of human behaviors. The more recent self-supervised algorithms remove this substantial limitation, and thus are substantially more plausible as potential candidate models of learning. What would have been a pointless and impossible comparison several years ago when all we had working for deep networks were supervised learning, has now become a comparison worth making.
>
>     Of course, just plausibility is just a first step -- and just an initial motivational idea. The whole point of our work here is to take the next step of actually checking whether given self-supervised algorithms actually do match human learning patterns, or not.  To the extent that our benchmarks quantify whether they do (or do not), we have gone beyond plausibility toward actual correctness evaluation.
>
> - *"line 162: The three conditions (Swapped, Non-Swapped, and Switch) model different learning environments and the amount of performance increase and decrease is different for each condition, so it would be meaningful to look at them separately, not averaging them. Is there a reason for averaging the results of all conditions?"*:
>
>     This is a good point. In the real-time benchmark, we average the three conditions as this averaged metric is already enough in differentiating the models. We indeed presented analyses on how each condition separately constrains the models. For example, the models with low initial $d'$ would make the Swap condition fail (see __SI__ Sec. 1.1.6 and Fig 8). The overlong replay window would make the Switch condition fail (see __SI__ Fig 9). These per-condition analyses show why some models fail, but a correct model for human learning should work in *all* conditions, which is what the averaged metric tries to capture and what the good models we have identified indeed do.  Ultimately, we think that the aggregated and disaggregated metrics are complementary for the real-time benchmark, and think it is useful to present both.

---

> ### Author Response · Authors · 2022-08-11
> **Response to the Correctness comments (2/3)**
>
>
> - *"line 177: If we want to see whether we can model human learning in temporal fashion through a life-long benchmark, shouldn't we see how the 10 evaluations of model performance change variates from human performance change? Here the authors averaged the results of 10 evaluations even though they contained temporally different evaluation results."*:
>
>     This is a great point. In the life-long learning benchmark, we use the averaged measure not only because it is a meaningful and useful metric for the AI community, but also because it is already strong enough to identify that existing models are *still significantly underperforming humans*  even on this simple measure. More specifically, even the best DNNs in our paper achieve only less than 0.4 accuracy on Mini-ImageNet. As Mini-ImageNet is a category-subsampled version of the full ImageNet, their performance on the full ImageNet is even lower than this number. But human adults are believed to perform better than 0.7 on ImageNet [1] (although only top-5 error is reported in [1], the corresponding DNN achieving similar top-5 error rate performs significantly better than 0.7 accuracy on ImageNet) and many properties of human vision are believed to be close to mature as early as 4-years-old [2]. Taken together, these facts suggest that our averaged measure seems like one reasonable approximation for the similarity to humans.
>
>     Moreover, the average metric is clearly sensitive enough to pick on important algorithmic differences -- for example, the key conclusion in our analysis work is mainly the total collapsing of the methods without negative samples in low batch-diversity settings, and this is clearly reflected in this averaged measure.
>
>     Finally, to support non-aggregated trajectory comparisons in the life-long-learning benchmark would require collecting the corresponding visual behavior trajectories on large numbers of infants -- an *extremely* difficult endeavor that would likely require many years of experimental work.  The reviewer's idea would be a great thing to try in theory -- but would *really* hard to do practice.
>
>     Given that we were able to use the averaged measure to extract both useful algorithm-to-algorithm comparisons, and a clear gap showing that even the best algorithms are far from ceiling, the averaged measure is a very effective compromise to allow something practical to be available for the next several years.  In the future if algorithms get good enough that average comparisons are no longer sufficient, and if the data for supporting them becomes available, that would be a great direction for future work.  (Though probably fairly far in the future.)
>
>     [1] Russakovsky, O., Deng, J., Su, H., Krause, J., Satheesh, S., Ma, S., ... \& Fei-Fei, L. (2015). Imagenet large scale visual recognition challenge. International journal of computer vision, 115(3), 211-252.
>
>     [2] Siu, C. R., \& Murphy, K. M. (2018). The development of human visual cortex and clinical implications. Eye and brain, 10, 25.
>
> - *"The authors passed too many details for the real-time benchmark to the prior work(Jia et al). ... For example, in a real-time benchmark, authors computed the categorization performance measured using d’(line 150), but they didn’t explain the definition of d’ and how to calculate it. Accordingly the next part, “to compute the learning effects through subtracting the changes of d’ on the exposed objects by the changes of d’ on the non-exposed objects” cannot be fully understood. Besides that, it would have been better if the authors explained that the exposure phase is the phase simulating unsupervised visual experience, and the reason for setting 3 experiment conditions (Swapped, Non-Swapped, and Switch)."*:
>
>     We have updated our paper and __SI__ to include more details for the real-time benchmark. We have also explained how $d'$ and the learning effect is computed (see __SI__ Sec. 1.1.3). The examples of exposure phase can be found in Fig 1 and __SI__ Fig 5. We use all three conditions as they are what human data were collected on, in the response above, we have also described how separate conditions are differently constraining the models.

---

> ### Author Response · Authors · 2022-08-11
> **Response to the Correctness comments (3/3)**
>
>
> - *"This paper focuses on using unsupervised algorithms to model human learning. I think it can be tested only with the real-time benchmark, so it seems unclear why the authors suggested the life-long learning benchmark. ..."*:
>
>     The trade-off figure in our paper (Fig 4) has shown that how two benchmarks are differently influenced by the choices of key parameters in our framework. So focusing only on one can miss important requirement for modeling human learning dynamics. For example, MAE works reasonably well on life-long, but not on real-time. MoCo v2 works pretty well on real-time, but worse than SimCLR on life-long. These two differentiation examples cannot be shown if just using one benchmark.
>     Moreover, the life-long learning benchmark also has direct utility for the AI community, which is not the case for the real-time learning benchmark. As we have argued in the previous response, the models are still underperforming humans in this measure, developing methods with better performance on this life-long benchmark is therefore not only what is needed to better model human learning dynamics, but also what is wanted by the AI community.
>
> - *"Imitating the way humans learn doesn't necessarily increase real-time and life-long learning performance of deep neural networks, then why is it important for DNN to achieve neural predictivity?"*:
>
>     Achieving neural predictivity by itself is important as it helps better understand the development of visual cortex in both real-time and life-long timescales. Moreover, the unsupervised DNNs trained only on human-perceivable data are still underperforming humans in many aspects, including the absolute performance on visual tasks (see above response) and other real-time learning tasks such as one-shot visual learning. So better modeling human learning dynamics should also help bridge this gap.
>
> - *"What is the reason for choosing Sam among 3 children in SAYCam?"*:
>
>     We select Sam as the number of video hours from this child is the longest among three children. Also, we have verified that the life-long learning results from another child, Alice, are highly consistent with the results from Sam (see __SI__ Fig 3). So we believe the choice of children does not influence our conclusions here.
>
> - *"line 161: Why using 4 test phases out of 5?"*:
>
>     We apologize for the confusion in our original description. Although there are 5 test phases, the measure from the first test phase, which is before the exposure phase, is used as the baseline measure in the learning effect computation (see __SI__ Sec. 1.1.3). So only the learning effects from the later four test phases are meaningful. We have updated our texts to make it clearer.

---

> ### Author Response · Authors · 2022-08-11
> **Response to the Additional Feedback**
>
>
> - *"If SAYCam shoots only for 2 hours every 2 weeks, we might miss many important events that happened during the weeks. ..."*:
>
>     Sam has 200 hours of video recordings in 2 years, so it's 2 hours per week by average, instead of 2 hours every 2 weeks. Indeed, this is still only a small portion of what infants actually perceive during their development. However, even using this small amount of data, the benchmark has already been useful in differentiating algorithms. As grouping multiple children or even using videos from adults may introduce undesired factors into the benchmark (see __SI__ Fig 3, where results from Ego4D expose the differences less strikingly than results from single child recordings), we believe we have done our best work in constructing this life-long benchmark. We welcome efforts in collecting datasets with even more hours. We would like to use datasets with longer video hours and have included this as one of our active plans for future work, but collecting such datasets requires years of work.
>
> - Other points:
>
>     We believe the updated paper and __SI__ have addressed the other points, including the numbers in tables and clearer explanation.

---

> ### Author Response · Authors · 2022-08-23
> **We are still looking forward to your response!**
>
> Dear Reviewer jYPN,
>
> As the discussion session will end in a week, we wonder whether you can kindly provide your further comments to our point-to-point response posted earlier and to our modified paper in the next several days so that we can have time to respond to them.
>
> Best,
> Authors

---

### Review · Ethics_Reviewer_piLC · 2022-08-23

**Recommendation:** 1

**Ethics Review:**

1 reviewer flagged the paper with etghical issue but it was because ethical and social implications were not discussed. Authors answered by including a paragraph about this. There is no major ethical issue with this paper otherwise....

---

> ### Author Response · Authors · 2022-08-27
> **Response**
>
> We thank reviewer piLC for reviewing our work and we are happy that this work is recommended as no serious ethical issues.

---

### Author Response · Authors · 2022-08-11
**General response**

We want to thank the reviewers for their hard work in providing feedback to this work.
Many reviewers mentioned the difficulty in understanding the real-time learning benchmark and the insufficient details in the Supplementary Information (__SI__) about it.
In response to this, we have updated both the main paper and the __SI__ to provide more introduction about this benchmark.
Moreover, we have added three pseudo-code algorithm descriptions about this benchmark, which we hope can make it even clearer.
Additionally, we have also provided the actual numbers for all algorithms tested in both the real-time and life-long learning benchmarks in the __SI__ (see Table 1 and 2), in response to comments from several reviewers.
The main paper is also updated with more detailed references to the tables, figures, and sections of the __SI__.

---

### Meta-Review · Area_Chair_xUxz · 2022-09-06

**Recommendation:** Accept
**Confidence:** 4

**Metareview:**

The two presented benchmarks for human-like real-time and life-long learning by incorporating perspectives from short-term and long-term knowledge acquisition and retention are highly interesting and useful for further research in the area of (unsupervised) human-like learning. The work is very well written and presented.

---

### Decision · Program_Chairs · 2022-09-16

Accept